# Improving the Performance of 2-To-4 Optical Decoders Based on Photonic Crystal Structures

**Mohammad Javad Maleki [1], Mohammad Soroosh [2],\*, and Ali Mir [1]**

[1]   Faculty of Engineering, Lorestan University, Khorramabad 68151-44316, Iran; maleki.mj@fe.lu.ac.ir (M.J.M.); mir.a@lu.ac.ir (A.M.)
[2]   Department of Electrical Engineering, Shahid Chamran University of Ahvaz, Ahvaz 61357-83135, Iran
\*   Correspondence: m.soroosh@scu.ac.ir; Tel.: +98-061-33330011-5641

**Abstract:** In this study, a novel, two-dimensional photonic crystal-based structure for the 2-to-4 optical decoder is presented. The structure consists of 23 rows and 14 columns of chalcogenide rods that are arranged in a square lattice with a spatial periodicity of 530 nm. The bias and the optical signals are guided toward the main waveguide through the three waveguides. Two unequal powers are applied to the input ports to approach the different intensities proportional to four working states into the main waveguide. Four cavities including the nonlinear rods are in response to drop the optical waves toward the output ports. To calculate the band diagram and the spatial distribution of the electric and magnetic fields, the plane wave expansion and the finite difference time domain methods have been used. The delay time of the designed structure is obtained around 220 fs, which is less than one for the previous structures. Furthermore, the gap between the margins for logic 0 and 1 is equal to 83%, which is higher than one for other works. Besides, the area of the structure is reduced to 90 $\mu m^2$ in comparison to all reported structures. Based on the mentioned results, it seems that an improvement of the performance for 2-to-4 optical decoders has been obtained in this research.

**Keywords:** optical decoder; optical Kerr effect; photonic bandgap; photonic crystal

## 1. Introduction

Optical systems have been proposed to obtain high-speed transmission and ultra-fast processing [1,2]. To use the great potential of the optical waves, all-optical systems should be designed and employed; so, designing all-optical devices has received great attention in the photonic field.

Photonic crystals (PCs) are the periodic arrays of the dielectric materials in one, two, and three dimensions [3,4]. They prepare a proper medium to control the flow of light. Due to their attractive effects, including the photonic bandgap (PBG) [5], scalability [6], slow light mode [7], and self-collimation [8], many attempts have been made for designing photonic crystal-based devices, such as decoders [9–16], encoders [17–19], analog-to-digital converters [20–22], adders [23–25], and flip-flops [26–28].

To guide a signal from a waveguide or line to another, the switching operation among the waveguides is required and it may be repeated several times for completing a defined task [29]. Optical decoders are known as the main blocks in optical integrated circuits. Some photonic crystal-based structures have been provided to all-optical 2-to-4 decoders [9–16]. Alipour et al. presented a square array of 29 and 22 rods, along with the x and z directions, where the lattice constant was equal to 825 nm [9]. The structure included three resonant rings to guide the optical waves toward the output ports. Although the structure correctly presented the decoding operation for two input ports, the time analysis was not completed, so the delay time and the difference of margins for logic 0 and 1 were not reported. Another structure based on five resonant rings for 2-to-4 decoders was proposed by

Mehdizadeh et al. [10]. They used chalcogenide rods in the form of 65 rows and 60 columns in which the spatial period was around 623 nm. In this report, the correct decoding operation was obtained; however, the time analysis was not completed. Based on a fundamental demultiplexer, they provided another structure that was composed of silicon rods [11]. It consisted of 89 rows and 51 columns, which were arranged with a lattice constant of 723 nm. To obtain the correct operation, they employed the inequal powers at the input ports in response to the different working states. Daghooghi et al. proposed a structure including six resonant rings made of chalcogenide rods [12]. The delay time of the structure was obtained around 6 ps and the area of the device was equal to 512 $\mu m^2$. The margins of logic 0 and 1 were calculated around 10% and 37% for digital applications. A structure based on four resonant rings was presented by Mehdizadeh et al. in which the silicon rods were placed in the form of 49 rows and 41 columns [13]. They used nonlinear rods in the rings to approach the optical Kerr effect and the dropping operation of optical waves toward the output ports. The area of the structure and the maximum frequency of switching were reported around 581 $\mu m^2$ and 10 GHz, respectively. Daghooghi et al. reduced the area of 2-to-4 optical decoders to 368 $\mu m^2$ in comparison to the previous works [14]. Using four resonant rings, they succeeded in increasing the mentioned gap up to 53% while the delay time was reported around 6.3 ps. Using three resonant rings, they modified the previous structure and decreased the delay time to 2 ps [15]. However, the area of the structure and the mentioned gap were changed to 380 $\mu m^2$ and 35%, respectively. Recently, they have succeeded in designing a PC-based 2-to-4 optical decoder, which is working at the slow light regime [16]. Based on using some defects and one resonant ring, they reported the delay time of 3 ps and the aforementioned gap of 78% for the presented decoder. Furthermore, the area of the structure was reported at around 228 $\mu m^2$.

When considering the mentioned structures for 2-to-4 optical decoders, it is revealed that the delay time and the area of the structure in addition to the amount of gap between the margins are taken into account as the main features for designing. In previous studies, the different trade-offs among them have generally been done so designing a structure that simultaneously improves them will be efficient work. In previous works, the dropping operation of optical waves is done by using the resonant rings, which result in a larger area and more delay time. In this study, a PC-based structure including four cavities has been proposed for optical 2-to-4 decoding operation. It consists of a doped glass for approaching the optical Kerr effect in the cavities. So, optical waves with different intensities have been allowed to drop toward the output ports corresponding to the different working states. The delay time of the presented structure is 220 fs, which is less than one for all previous works [12,14–16]. Furthermore, the area of the structure has been reduced to 90 $\mu m^2$ in comparison to one for the previous structures [9–16]. Besides, the gap between the margins for the proposed decoder is around 83%, which is higher than the obtained result in the last reports [12,14–16]. As a result, the designed decoder shows higher performance than other decoders, and it can be considered as the proper device in optical processing applications.

In the next section, the designed structure is introduced and then the obtained results of the simulation are presented in Section 3. The conclusion of this work will be expressed in Section 4.

## 2. The Proposed Structure for the 2-To-4 Decoder

The fundamental structure consists of a two-dimensional array of chalcogenide rods that are arranged in a square lattice. A number of 23 and 14 rods with the air gap along the x and z directions are assumed in the form of the lattice with a spatial period (a) of 530 nm. The refractive index of the dielectric rods is assumed to be 3.1 [30]. By choosing a = 530 nm, the gap map of the structure was calculated for both transverse electric (TE) and transverse magnetic (TM) polarizations as shown in Figure 1a. The bandgap of the photonic crystal-based structure is a crucial factor for guiding the light and should be calculated at the first step of the designing. To obtain the bandgaps, one should calculate the band diagram of the structure. In this study, the plane wave expansion (PWE) method with the perfectly matched layer (PML) condition was used to calculate the band diagram, as shown in

Figure 1b [31]. The Brillouin zone and a top view of the rods are presented in Figure 1b. In this method, the components of both electric and magnetic fields were expanded in terms of the Fourier series components along the reciprocal lattice vector, and the obtained eigenvalue relations were solved. By choosing a radius of 106 nm for the fundamental rods, it can be seen that the width of the photonic bandgap is equal to 0.126, which is large enough. In this case, $a/\lambda = 0.342$ is approximately placed at the center of the photonic bandgap, where $\lambda$ is the wavelength. In respect to a = 530 nm, a wavelength of 1550 nm has been obtained for incoming optical waves. As shown in Figure 1a, the blue and red colors present the bandgaps at TE and TM modes, respectively. It can be seen that the vertical dashed line that is proportional to a radius of 106 nm for the fundamental rods does not cross the TM bandgap. This figure demonstrates that there is no photonic bandgap at TM mode. It can be seen that the mentioned array covers one bandgap at $0.31 \leq a/\lambda \leq 0.436$ for TE polarization, so the photonic bandgap is obtained at 1215 nm $\leq \lambda \leq$ 1710 nm. Since the third optical communication window is included in this interval, the proposed structure can be matched to the optical systems. The wavelength that is set in the PBG will be allowed to propagate throughout the structure through the waveguides.

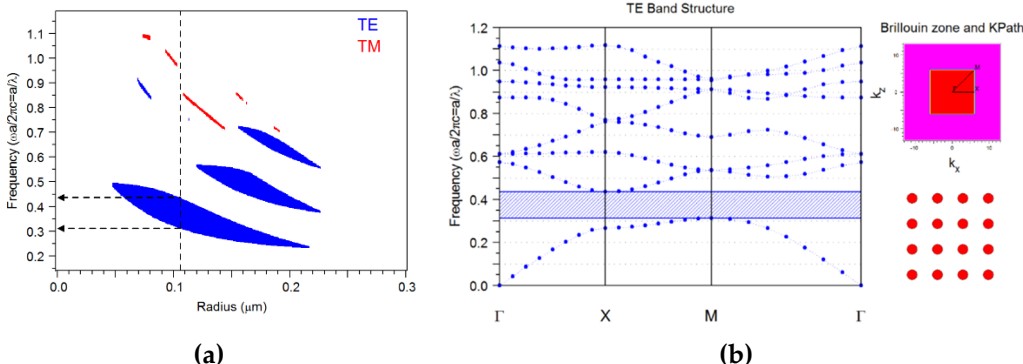

**Figure 1.** (**a**) The calculated gap map and (**b**) the band diagram with the Brillouin zone of the fundamental structure. The red circles show chalcogenide rods in the form of the square lattice.

The proposed structure for decoding operation is constructed by eight waveguides, which are connecting two input ports, X and Y to four output ports, O0, O1, O2, and O3 (see Figure 2a). To create a waveguide, one row (or column) of rods was removed. Due to the periodicity of the fundamental structure, each rod (or column) with an air gap to the adjacent rod can be assumed as a pair layer, which is alternatively repeated. Based on the distributed Bragg reflectors, to prevent the interference of two adjacent waveguides, at least four rows (or columns) of rods should be generally assumed between them. In respect to this issue, it has been tried that the least number of rods has been used in designing the structure, so 14 rows and 23 columns of rods have been considered. One bias signal from port E is applied to the waveguide W1 for interfering with two signals coming from input ports through the waveguides W2 and W3. The resulting signal is guided toward the waveguide W4. In response to the amount of the optical intensity in W4, four waveguides W5, W6, W7, and W8 drop the waves toward the output ports O0, O1, O2, and O3, respectively. Four nonlinear rods made of a doped glass with a linear refractive index of 1.4 and a nonlinear coefficient of $10^{-14}$ m$^2$/W are placed in waveguides W5, W6, W7, and W8 [32]. Due to the optical Kerr effect, the value of the refractive index for a dielectric material (n) depends on the applied optical intensity (I). This effect is generally defined as $n(I) = n_1 + n_2 \times I$, where $n_1$ is the linear refractive index and $n_2$ is the nonlinear coefficient [33,34]. According to Bragg's theory, the incident waves with the wavelength of $\lambda$ and the reflected waves from the periodic bilayers are in phase if the equation $n_a d_a + n_b d_b = \lambda/2$ is satisfied, where $n_a$ and $n_b$ are the refractive indices and $d_a$ and $d_b$ are the thickness of two layers [35]. Considering the square lattice of the dielectric rods in the air gap, one rod and air gap are assumed as two different layers for the mentioned equation. To obtain the dropping operation at the same wavelength for the cavities, the left side of the mentioned equation should be kept in a constant value. So, changing the radii of the

nonlinear rods assists to satisfy the equation. As a result, using the different radii in the cavities, in addition to the optical Kerr effect, makes the dropping operations for the different intensities through the cavities. By changing the values of radii for nonlinear rods with a step of 1 nm and calculation of the power at output ports, the radii of 118 nm, 111 nm, 104 nm, and 97 nm were assumed for rods with green, yellow, blue, and black colors in waveguides W5, W6, W7, and W8, respectively. So, the different optical intensities into waveguide W4 in response to the different states of the input ports will be dropped through waveguides W5, W6, W7, and W8 for achieving the decoding operation. The radii of some rods in waveguides W1, W2, and W3 have been given in Figure 2b. These rods, like the fundamental ones, have been shown with red color and are made of the chalcogenide. In the corners of waveguides W2 and W3, two rods with the radii of 64 nm have been placed to increase the transmission of optical waves at the bends. Four rods with the radii of 74 nm, in addition to one rod with a radius of 70 nm, are considered in waveguides W1, W2, and W3 for approaching the desired interferences at the cross-connections of the waveguides. The values of the mentioned rods have been obtained by scanning the different radii and calculating the transmission factor. The structural parameters of the presented decoder are summarized in Table 1.

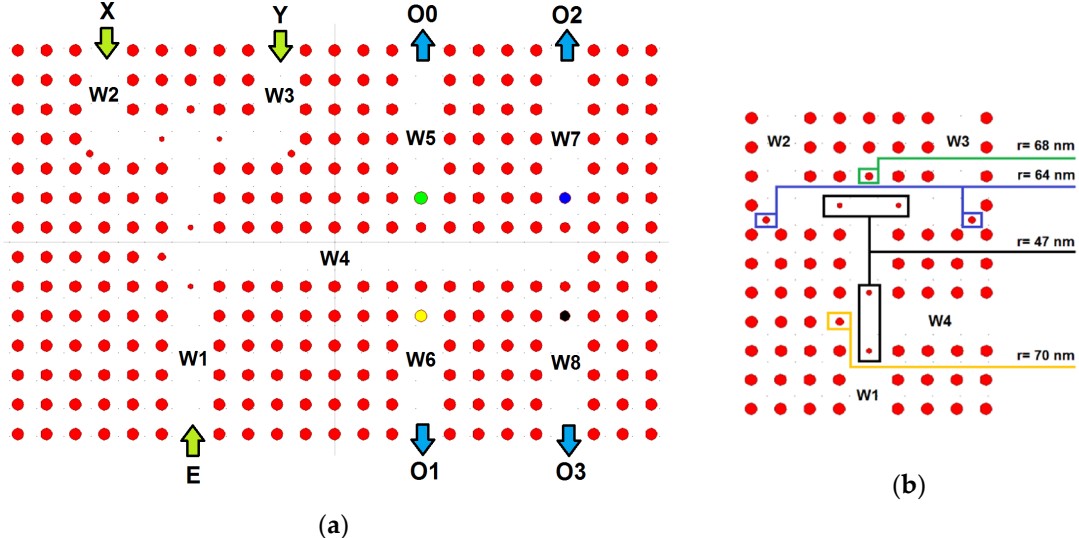

**Figure 2.** (**a**) The proposed structure for the optical 2-to-4 decoder and (**b**) presentation of the defects in waveguides W1, W2, and W3.

**Table 1.** Structural parameters of the proposed decoder.

| Parameter | Value | Unit |
| --- | --- | --- |
| Lattice constant | 530 | nm |
| Linear refractive index of the chalcogenide | 3.1 | - |
| Linear refractive index of the doped glass | 1.4 | - |
| Nonlinear refractive index of the chalcogenide | $1 \times 10^{-20}$ | m$^2$/W |
| Nonlinear refractive index of the doped glass | $1 \times 10^{-14}$ | m$^2$/W |
| Radius of fundamental rods | 106 | nm |
| Radius of the nonlinear rod in W5 | 118 | nm |
| Radius of the nonlinear rod in W6 | 111 | nm |
| Radius of the nonlinear rod in W7 | 104 | nm |
| Radius of the nonlinear rod in W8 | 97 | nm |

## 3. Simulations and Discussions

To simulate the optical wave throughout the structure, the finite difference time domain method was used. The electric and magnetic fields were calculated in both space and time domains. The length of unit cells at x ($\Delta x$) and z ($\Delta z$) directions should be less than $\lambda/10$ and the time step ($\Delta t$) should satisfy

the inequality $c\Delta t < 2^{-0.5}\Delta x$ for case $\Delta x = \Delta z$, where c is the speed of light in the vacuum [36]. In this study, $\Delta x = \Delta z = 100$ nm and $\Delta t = 0.1$ fs were used to calculate the components of the fields.

As mentioned in Section 2, the proposed structure includes two inputs, X and Y, so four working states should be considered. In the first state, for X = Y = 0, one optical signal with the intensity of I0 is applied through the bias port E. The optical waves reach the waveguide W4 and the waveguide W5 drops them toward the output port O0 because the cavity including the green rod is set to the optical intensity of I0. As a result, the output port O0 will be ON, as shown in Figure 3a. The second state is assumed for X = 1 and Y = 0, where the waveguide W1, in addition to the waveguide W3, guides the waves toward the waveguide W4. Since the incoming optical intensity from the port X is equal to I0, the approximate intensity of 2I0 reaches the waveguide W4. In this case, the waveguide W6 transmits the waves toward the port O1 (Figure 3b). For the third state, the applied optical intensity to the port Y is assumed to be 2I0 while the port X is inactive (X = 0 and Y = 1). So, the intensity of 3 I0 is propagated into the waveguide W4 and then waveguide W7 guides the waves toward the port O2 (Figure 3c). In the last state, due to the applied optical signals to both input ports (X = Y = 1), the optical intensity of 4I0 will reach that of the waveguide W4. In response to the mentioned intensity, the waveguide W8 guides the optical waves toward the output O3 (Figure 3d).

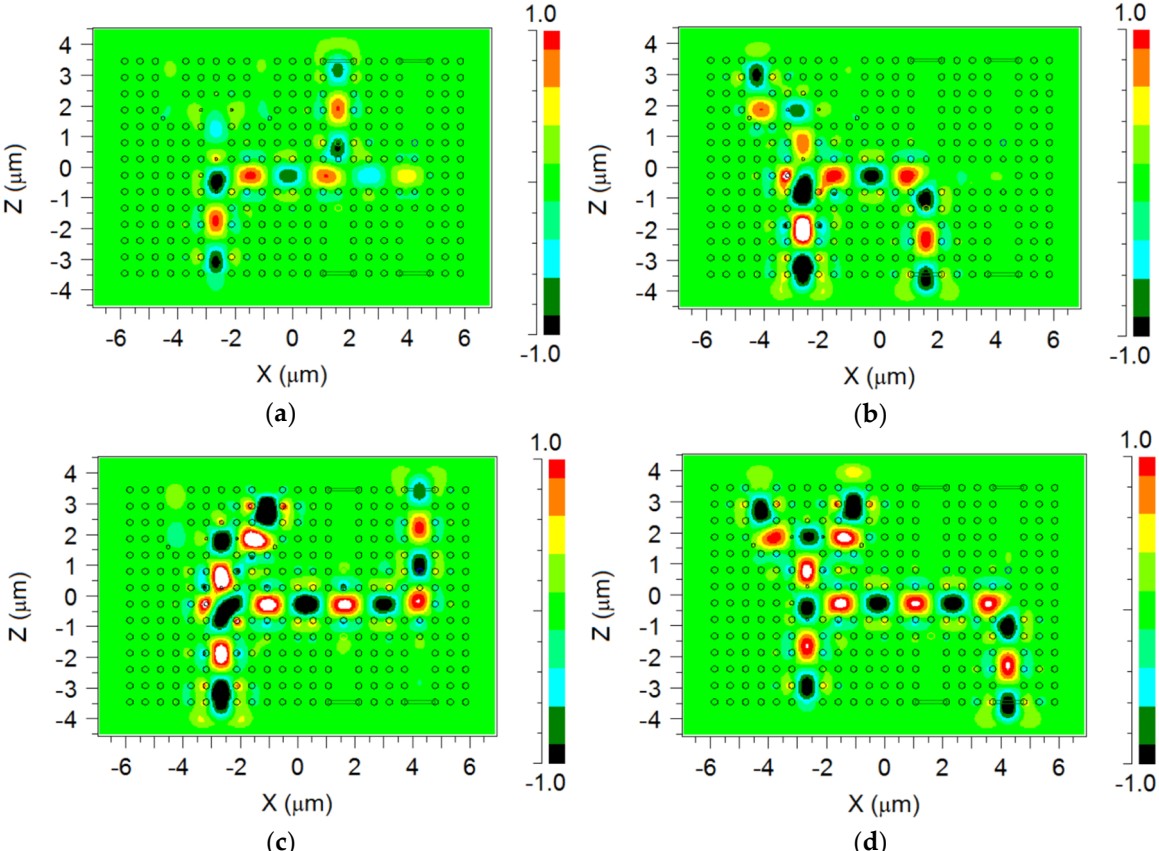

**Figure 3.** The optical wave propagation throughout the proposed decoder for states: (**a**) X = Y = 0, (**b**) X = 1, Y = 0, (**c**) X = 0, Y = 1, and (**d**) X = Y = 1.

To further evaluate the dropping operation, the different intensities were applied to the waveguide W4 and the normalized powers at output ports were calculated. Figure 4 demonstrates that the maximum dropping through the waveguides W5, W6, W7, and W8 will occur approximately for I0, 2I0, 3I0, and 4I0, respectively, where I0 is equal to 10 mW/$\mu m^2$.

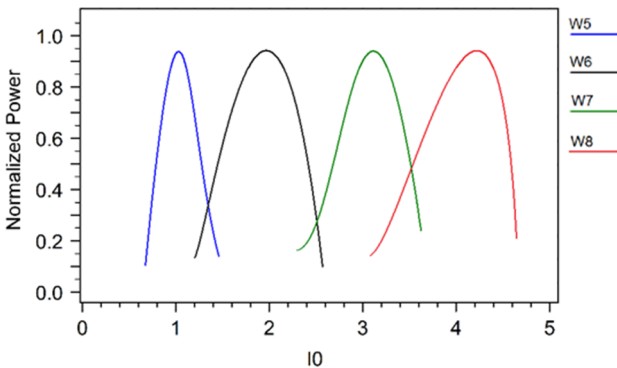

**Figure 4.** The normalized power at waveguides W5, W6, W7, and W8 versus the optical intensity I0. The amount of I0 is 10 mW/μm$^2$.

Although the correct operation of the proposed decoder has been shown in Figure 3, the time analysis of the structure must be considered for all possible states. To do this, the optical pulses with a duration of 3 ps are applied to the input ports. So, the normalized powers at output ports versus the time have been shown in Figure 5 for different working states. It can be inferred that the maximum fall time of the proposed is equal to 145 fs. The fall time is defined as the required time to a signal fall 90% of the initial value. Furthermore, the maximum steady-state time in the error band 2% is obtained to be 1.9 ps.

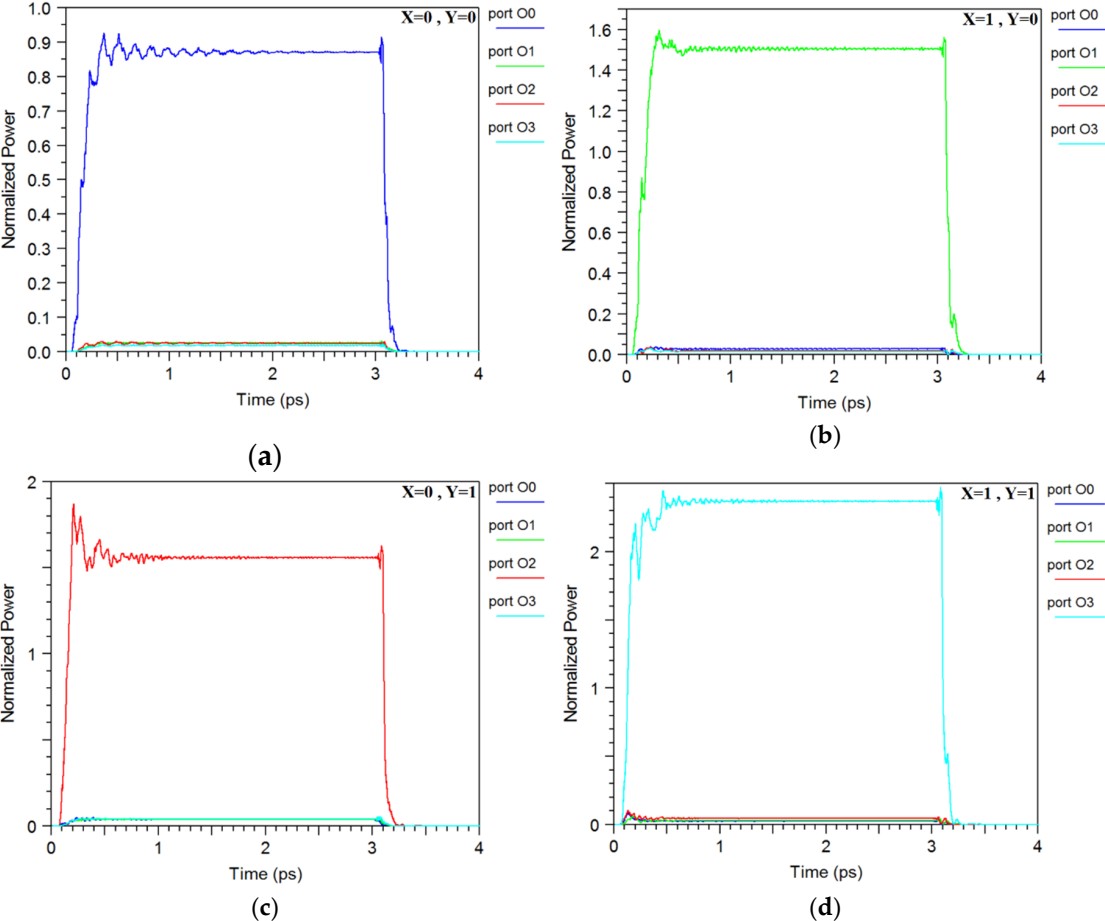

**Figure 5.** The time response of the proposed structure for different states (**a**) X = Y = 0, (**b**) X = 1, Y = 0, (**c**) X = 0, Y = 1, and (**d**) X = Y = 1.

The details of the time analysis for more assessment have been presented in Table 2. The normalized powers in addition to the assigned logics at the ports O0, O1, O2, and O3 are given for the possible states. The highest steady-state value of logic 0 is around 4% for case X = Y = 1. Furthermore, the lowest value of the normalized power for logic 1 is proportional to the case X = Y = 0, in which it is equal to 87%. As a result, the margins of the structure for logic 0 and 1 are assumed to be 4% and 87%, respectively. The delay time is defined as the required time for a signal to reach 90% of its final value. It can be seen that the delay time of the structure is equal to 220 fs. The fall time is approximately obtained at 145 fs for state X = Y = 1. Here, the time is the duration of one that falls in a 10% initial value. Furthermore, the maximum steady-state time in error band 2% is obtained to be 1.9 ps.

**Table 2.** The obtained results of the time analysis.

| Inputs | | Output Ports | | | | | | | | Delay Time (fs) | Fall Time (fs) | Steady-State Time (ps) |
|---|---|---|---|---|---|---|---|---|---|---|---|---|
| | | Logic | | | | Normalized Power (%) | | | | | | |
| X | Y | O0 | O1 | O2 | O3 | O0 | O1 | O2 | O3 | | | |
| 0 | 0 | 1 | 0 | 0 | 0 | 87 | 3 | 3 | 2 | 200 | 130 | 1.9 |
| 1 | 0 | 0 | 1 | 0 | 0 | 3 | 151 | 3 | 2 | 210 | 125 | 1.6 |
| 0 | 1 | 0 | 0 | 1 | 0 | 3 | 3 | 155 | 3 | 205 | 120 | 1.7 |
| 1 | 1 | 0 | 0 | 0 | 1 | 3 | 4 | 4 | 239 | 220 | 145 | 1.6 |

Although it seems that the obtained results are proper to be considered for optical applications, the comparison of them with ones for other works reveals the performance of the proposed structure as tabulated in Table 3. The calculated delay time has been presented in the second column. One can see that the proposed structure is faster than all designed structures. This issue is an important advantage of considering optical processing applications. According to the third column, the area of the structure used in this work is smaller than one for all previous works. In this study, the area of the all-optical 2-to-4 decoder has been reduced to 90 $\mu m^2$ in compared to one for references [9–16]. Furthermore, as shown in the fourth column, the difference between the aforementioned margins (DM) has been obtained around 83%, which is higher than that for the references [12,14–16]. The last column demonstrates that the required optical intensity in this work is lower than one for all previous works. It can be concluded that the compactness, ultra-fast response, low power operation, and a large amount of the gap between logic 0 and 1 are the main advantages of the designed structure.

**Table 3.** The comparison of the obtained results of this work with ones for other works.

| Work | Delay Time (ps) | Area ($\mu m^2$) | DM (%) | I (mW/$\mu m^2$) |
|---|---|---|---|---|
| [9] | - | 434 | - | 100 |
| [10] | - | 1513 | - | 200 |
| [11] | - | 2373 | - | 20 |
| [12] | 6 | 184 | 27 | - |
| [13] | - | 581 | - | 50 |
| [14] | 6.3 | 368 | 53 | 13 |
| [15] | 2 | 380 | 35 | 100 |
| [16] | 3 | 228 | 78 | 13 |
| This work | 0.22 | 90 | 83 | 10 |

## 4. Conclusions

In this study, a two-dimensional photonic crystal-based structure for all-optical 2-to-4 decoders has been presented. The structure included 23 columns and 14 rows of chalcogenide rods with a square arrangement, in which the lattice constant was equal to 530 nm. One bias and two input signals were guided to the main waveguide of the device. Due to using the nonlinear rods with the different radii in four cavities, the dropping operations for the optical intensity of I0, 2I0, 3I0, and 4I0, were

obtained through four waveguides connected to the main waveguide. The value of I0 was adjusted to 10 mW/$\mu$m$^2$. The delay time of the proposed structure was obtained around 220 fs, which proved the presented device was faster than all reported structures. As another advantage, the area of the structure was decreased to 90 $\mu$m$^2$ in comparison to previous works. Furthermore, the difference of the margins for logic 0 and 1 was calculated at around 83%. In respect to the obtained results, it can be concluded that the proposed decoder can be considered in optical integrated circuits.

**Author Contributions:** Formal analysis, M.J.M. and M.S.; Methodology, M.J.M. and M.S.; Writing-Original Draft, M.J.M.; Writing-Review and editing, M.S. and A.M.

**Funding:** This research received no external funding.

**Conflicts of Interest:** The authors declare no competing financial interest.

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
