# Peer review of "Improving the Performance of 2-To-4 Optical Decoders Based on Photonic Crystal Structures"

_crystals, doi:10.3390/cryst9120635_

Round 1
Reviewer 1 Report
The paper reports about the two-dimensional photonic crystal-based structure for a 2-to-4 optical decoder. The structure consists of the 23 rows and 14 columns of chalcogenide rods that are arranged in a square lattice with the spatial periodicity of 530 nm. The band diagram and the spatial distribution of electric and magnetic field was obtain by FDTD method. Author show that the delay time of the designed structure was obtained around 220 fs that is less than one for the previous reports. Furthermore, the calculated gap between logic 0 and 1 was equal to 83% that is less than one for other works. Besides, the area of the structure was reduced to 90 µm2 in compared to all reported structures.
The manuscript is well structured, nicely written and I consider it is suitable for the publication after minor revision.
I have the following suggestions and comments:
Comment 1: In the first figure, the authors present the band structures of PC, but do not present a sketch view of the structure. The sketch view should be added in the Fig.1a
Comment 2: This calculation work and reproducibility of the presented results is important for it. The authors presented the model in detail, but not all its parameters are given in the text. For example, the text does not specify the radii of the rods marked in green (see Fig.)

Reviewer 2 Report
This paper presents a 2x4 optical decoder based on Photonic crystals waveguides and the Kerr nonlinearity of a material. It is a different approach to previous results obtained by the authors since it does not include any ring resonators. The subject of the paper is of reduced interest, since the design of optical decoders is not a very active research area and no experimental data are given. However, the proposed design obtains better characteristics in terms of delay time, area and power contrast than previous devices and can hence be considered for publication. Some points should be addressed in order to improve the manuscript. Specifically
1.- English use, although not as bad as to prevent the reader from understanding the content, should be improved.
2.- In the last line of the second paragraph the references for the adders should be [23-25].
3.- More details on the design decisions should be given:
3a.-Specifically, the authors should make some comments on why they choose a rod radius of 106 nm, and 23 and 14 rods in the x and z directions, i.e, how the rod radius and number of rods affect the design of the waveguides.
3b.- It is also very important to specify the material used for the nonlinear rods, the authors only say that they are made with doped glass, with no further indication on the doping. They send the author to reference [22] but in that work there are not full specifications (if it is a secondary reference, please use the original reference). Moreover, in [22] all the rods are made of silicon, but in the present case the other rods are made of chalcogenide (no indication on the exact chalcogenide used either). This change of material could make the fabrication of the structure almost unfeasible.
4.- Only the TE band diagram is given. Does the TM modes also present a bandgap in 1550 nm? This aspect and its impact on the device performance needs to be explained.
5.-In figure 2, in the bending of waveguides W2 and W3 (in the Y-structure) and the bending form W1 o W4 some rods smaller than the rest and in some cases in non regular positions are drawn, but the authors do not mention them in their paper. It is essential that they explain why these especial rods are needed and how they are designed.
6.-The authors simulate the time response of the device when a code is introduced in the input (figure 4), it could be interesting to simulate as well how it reacts when the code is “switched of” . They obtain the delay time as the time for a signal to reach the 90% of its final value. Along with this parameter, the “settling time”, i.e, the time for the signal to reach and remain within a given error band is also and important parameter that the authors should include in their tables.
7.- One of the most important drawbacks of a non-linear device is the amount of power that it requires to operate. The paper only gives normalized power, but a figure on the needed powers should also be indicated and a comment on this issue or comparison with other works regarding this aspect included. Moreover, the authors do not explain how to specify the nonlinear rods radius so that the waveguides are permitted or not. A clearer explanation on the working principle is clearly missing.
Round 2
Reviewer 2 Report
Most of my previous comments have been taken into consideration, but there are still some points to improve.
The authors say that have made a complete revision of the English use in the manuscript, but there are many mistakes that result in an unconfortable reading of the paper.
The authors give results, Fig. 3 and 4, that confirm that guiding through the output waveguides is controlled by the intensity in another point of the structure, resulting in the correct operation of the decoder. However, they fail to explain the underlying concept that makes the device behave as desired; they state that it is based on the Kerr effect, but they do not explain how different radii of the rods affect this nonlinear effect
